# Fooling a Complete Neural Network Verifier

**Dániel Zombori, Balázs Bánhelyi, Tibor Csendes, István Megyeri, Márk Jelasity**
Institute of Informatics, University of Szeged, Hungary
`{zomborid, banhelyi, csendes, imegyeri, jelasity}@inf.u-szeged.hu`

## Abstract

The efficient and accurate characterization of the robustness of neural networks to input perturbation is an important open problem. Many approaches exist including heuristic and exact (or complete) methods. Complete methods are expensive but their mathematical formulation guarantees that they provide exact robustness metrics. However, this guarantee is valid only if we assume that the verified network applies arbitrary-precision arithmetic and the verifier is reliable. In practice, however, both the networks and the verifiers apply limited-precision floating point arithmetic. In this paper, we show that numerical roundoff errors can be exploited to craft adversarial networks, in which the actual robustness and the robustness computed by a state-of-the-art complete verifier radically differ. We also show that such adversarial networks can be used to insert a backdoor into any network in such a way that the backdoor is completely missed by the verifier. The attack is easy to detect in its naive form but, as we show, the adversarial network can be transformed to make its detection less trivial. We offer a simple defense against our particular attack based on adding a very small perturbation to the network weights. However, our conjecture is that other numerical attacks are possible, and exact verification has to take into account all the details of the computation executed by the verified networks, which makes the problem significantly harder.

## 1 Introduction

In their seminal work, Szegedy et al. found that for a given neural network and input example one can always find a very small adversarial input perturbation that results in an incorrect output (Szegedy et al., 2014). This striking discovery motivated a substantial amount of research. In this area, an important research direction is *verification*, that is, the characterization of the robustness of a given network in a principled manner. A usual way of defining the verification problem involves the specification of an input domain and a property that should hold over the entire domain. For example, we might require that all the points within a certain distance from an input example share the same output label as the example itself. The verification problem is then to prove or disprove the property over the domain for a given network (Bunel et al., 2020).

There are a large number of verifiers offering different types of guarantees about their output. Complete verifiers offer the strongest guarantee: they are able to decide whether a given property holds in any given input domain. For example, the verifier of Tjeng et al. is a state-of-the-art complete verifier that we will focus on in this paper (Tjeng et al., 2019). However, it is currently standard practice to ignore the details of the computations that the network under investigation performs, such as the floating point representation or the order in which input signals are summed.

In this paper, we claim that such implicit assumptions make verifiers vulnerable to a new kind of attack where the attacker designs a network that fools the verifier, exploiting the differences between how the verifier models the computation and how the computation is actually performed in the network. We will argue that such attacks can achieve an arbitrary divergence between the modeled and the actual behavior.

This new attack has practical implications as well. Concerns about the safety of AI systems are expected to lead to the establishment of standard requirements certified by a designated authority (Salis-Madinier, 2019). These certification procedures might involve verification methods as well. Fooling such methods makes it possible to get unsafe systems certified that might even contain a backdoor allowing for triggering arbitrary behavior.

Numerical precision has not been a key practical concern in machine learning. Networks do sometimes produce numerical errors (e.g., Inf or NaN values), most often due to the non-linear operations within the loss function (Odena et al., 2019) or divergence during training. However, the network weights are normally robust to small perturbations due to stochastic learning algorithms (Bottou, 2010), and due to regularizers such as standard variants of weight decay and dropout (Srivastava et al., 2014). Due to this robustness, low precision arithmetic can be applied as well (Courbariaux et al., 2015; Gupta et al., 2015). Our results indicate that, when it comes to exact methods for verification, *numerical issues become a central problem that can cause arbitrary errors and enable backdoors*.

Our contributions are the following. In Section 3, we introduce a simple adversarial network that misleads the verifier of Tjeng et al. (2019). In Section 4, we show how to hide the large weights that are present in the simple network. In Section 5, we describe a way to add a backdoor to an existing network with the help of the adversarial networks we proposed. Finally, in Section 6 we offer a defense against the attack we presented.

## 2 BACKGROUND

Let us first formulate the verification problem, namely the problem of checking whether a given property holds in a given domain. We adopt the notation used in (Tjeng et al., 2019). For a possible input $x$, let $\mathcal{G}(x)$ denote the set of inputs that are considered similar to $x$ in the sense that we expect all the points in $\mathcal{G}(x)$ to get the same label as $x$. The set $\mathcal{G}(x)$ is normally defined as a ball around $x$ in some metric space defined by a suitable vector norm. The input domain we need to consider is given as $\mathcal{G}(x) \cap \mathcal{X}_{valid}$ where $\mathcal{X}_{valid}$ denotes the valid input points. For example, we have $\mathcal{X}_{valid} = [0, 1]^m$ if the input is an image of $m$ pixels with each pixel taking values from the interval $[0, 1]$.

We now have to formulate the property that we wish to have in this domain. Informally, we want all the points in the domain $\mathcal{G}(x) \cap \mathcal{X}_{valid}$ to get the same classification label as $x$. Let $\lambda(x)$ denote the true label of $x$ and let $f(x; \theta) : \mathbb{R}^m \to \mathbb{R}^n$ denote the neural network, parameterized by $\theta$. This network has $n$ outputs classifying each input $x$ into $n$ classes. The label of $x$ as predicted by the network is given by $\arg\max_i f(x; \theta)_i$. Using this notation, the property we wish to have for an input $x' \in (\mathcal{G}(x) \cap \mathcal{X}_{valid})$ is that $\lambda(x) = \arg\max_i f(x'; \theta)_i$.

Putting it all together, the verification problem can be expressed as deciding the feasibility of the constraint

$$x' \in (\mathcal{G}(x) \cap \mathcal{X}_{valid}) \wedge (\lambda(x) \neq \arg\max_i f(x'; \theta)_i), \tag{1}$$

with $x'$ as our variable. If this constraint is feasible then there is an $x'$ that violates the property. If it is infeasible then (provided $\mathcal{G}(x) \cap \mathcal{X}_{valid}$ is not empty) there is no such $x'$.

### 2.1 APPROACHES TO VERIFICATION

There are many approaches to tackle this problem. We can, for example, search for a suitable $x'$ in the given domain using some heuristic optimization methods (Goodfellow et al., 2015; Moosavi-Dezfooli et al., 2016; Kurakin et al., 2017; Carlini & Wagner, 2017; Brendel et al., 2019). If the search succeeds, we can decide that equation 1 is feasible. Otherwise we cannot decide.

Other methods attempt to find a proof for the infeasibility of equation 1, however, they do not guarantee such a proof. Examples include (Wong & Kolter, 2018; Weng et al., 2018; Gehr et al., 2018; Raghunathan et al., 2018; Singh et al., 2019). If a proof is found, we can decide that equation 1 is infeasible. Otherwise we cannot decide. Such methods are sometimes called *incomplete* (Tjeng et al., 2019; Bunel et al., 2020).

The strongest guarantee is given by methods that are able to decide the feasibility of equation 1. These methods are sometimes called *complete* (Tjeng et al., 2019; Bunel et al., 2020).

Examples for such methods include Reluplex (Katz et al., 2017), a method based on an SMT solver. A number of verifiers are based on MILP solvers, for example, (Cheng et al., 2017; Dutta et al., 2018). MIPVerify (Tjeng et al., 2019) also uses an MILP formulation along with several additional techniques to improve efficiency (see Section 2.2). Symbolic interval propagation has also been proposed for ReLU networks by Wang et al. in ReluVal (Wang et al., 2018b), and as part of Neurify (Wang et al., 2018a). In Neurify, interval propagation is used as a technique to tighten the

bounds used for linear relaxation. Nnenum is another geometric method that is based on propagating linear star sets (Bak et al., 2020).

## 2.2 MIPVERIFY

Although the idea behind the attack is not specific to a particular verifier—as we discuss in Section C of the Appendix—we develop and evaluate the attack in detail for a state-of-the-art complete verifier: MIPVerify (Tjeng et al., 2019). It is based on a mixed integer linear programming (MILP) formulation. As long as the domain $\mathcal{G}(x) \cap \mathcal{X}_{valid}$ is the union of a set of polyhedra, and the neural network $f(x, \theta)$ is a piecewise linear function of $x$ with parameters $\theta$, the problem of checking the feasibility of the constraint in equation 1 can be formulated as a MILP instance.

$\mathcal{G}(x)$ is normally defined as a ball in a suitable norm with $x$ as the center. In $\ell_\infty$ or $\ell_1$ norms $\mathcal{G}(x)$ is thus a cube. Also, $\mathcal{X}_{valid}$ is normally a box or a set of boxes, so the domain is indeed the union of a set of polyhedra. The neural network is piecewise linear as long as the nonlinearities used are ReLUs (note that the last softmax normalization layer adds no extra information and can thus be ignored). For the details of the MILP formalization, please see (Tjeng et al., 2019).

Importantly, MIPVerify applies a presolve step that greatly increases its efficiency. In this step, the authors attempt to tighten the bounds on the variables of the problem, including on the inputs to each ReLU computation. If in this step it turns out that the input of a ReLU gate is always non-positive, the output can be fixed as a constant zero, and if the input is always non-negative then the ReLU gate can be removed from the model as it will have no effect.

The presolve step applies three approaches in a progressive manner. First, a fast but inaccurate interval arithmetic approach is used. The resulting bounds are further improved by solving a relaxed LP problem on every variable. Finally, the full MILP problem is solved for the variables but with early stopping.

## 2.3 FLOATING POINT REPRESENTATION

Floating point real number representations are successful and efficient tools for most real life applications (Muller et al., 2010). This arithmetic is available on most modern computers via sophisticated hardware implementations. A floating point number is represented as $s \cdot b^e$, where $s$ is the signed *significand*, $b$ is the *base* and $e$ is the *exponent*. There are numerous standards to implement the exact details of this idea that differ mainly in the number of bits that the significand and the exponent use. The formula to compute the represented real number has several possible variations as well.

Here, we will use the double precision (binary64) arithmetic defined by the IEEE 754-1985 standard (IEEE, 1985). There, $b = 2$ and we have a sign bit, an 11 bit exponent, and a 53 bit significand (with 52 bits stored explicitly). The so called machine epsilon (the maximum relative rounding error) is $2^{-53} \approx 1.11e-16$. This means that, for example, the computation $10^{20} + 2020 - 10^{20}$ will result in zero in this representation, if executed in the specified order. In our attack, we will exploit roundoff errors of this type. Note, that in the order of $10^{20} - 10^{20} + 2020$ we obtain the correct result of 2020.

## 2.4 WHAT IS THE OBJECT OF VERIFICATION?

In related work, it is almost always implicitly assumed that the object of verification is a neural network computed using *precise arithmetic*. However, a more appropriate and also more ambitious goal is to consider the network as it is computed in practice, that is, using *floating point arithmetic*, with an *arbitrary ordering* of the parallelizable or associative operations.

If we consider the less ambitious goal of the verification of the precise model, most complete methods still fall short as they use floating point representation internally without any hard guarantees for precision. Those verifiers that are based on different linear programming formulations all belong to this category. Although Katz et al. explicitly consider this issue for Reluplex (Katz et al., 2017), they also propose floating point representation citing efficiency reasons.

However, striving for precision is a wrong direction as actual networks use floating point representation themselves. This fact means that actual networks include non-linearities that need to be modeled

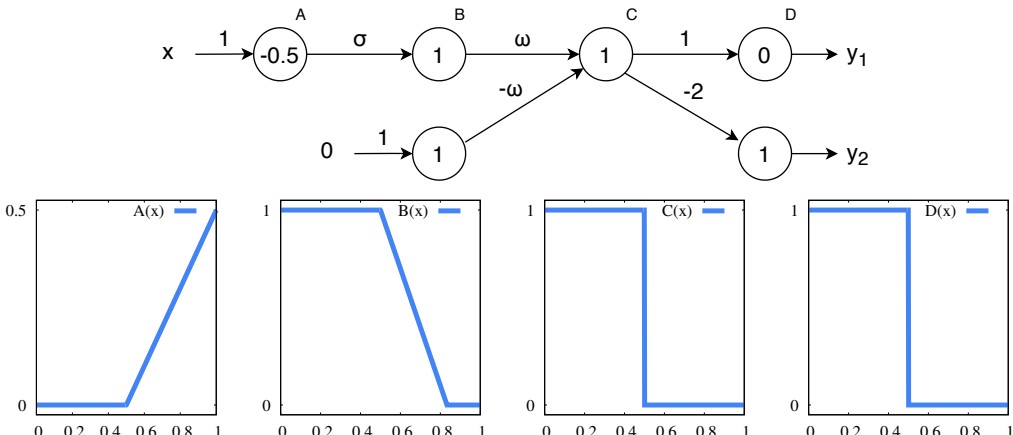

Figure 1: A naive adversarial network. The neurons (represented by circles) are ReLUs, the numbers in the circles represent the additive bias, and the numbers over the connection represent the weight. The valid input range is assumed to be $x \in [0, 1]$. Parameter $\sigma$ defines the steepness of the transition region in $B(x)$, while $\omega$ is the "large weight" parameter. The output of four different neurons is shown as a function of $x$, assuming $\sigma = -3$ and $\omega = 10^{17}$.

explicitly if our goal is verification with mathematical strength. For example, ReluVal (Wang et al., 2018b) is a promising candidate to meet this challenge. It is a symbolic interval-based method that attempts to compute reliable lower and upper bounds for the network activations in a ReLU network. Unfortunately, when computing the parameters of the linear expressions for the symbolic intervals, it still uses a floating point representation, which means the method is not completely reliable in its published form. Similarly, Nnenum is also deliberately implemented in an unreliable manner due to efficiency Bak et al. (2020).

Our main point here is that any "sloppiness" in the definition of the object of verification or cutting any corners for the sake of efficiency are potential sources of security problems that compromise the reliability of verification. Here, we shall give an example of a successful attack on MIPVerify. We will exploit the fact that—even if both the network and MIPVerify use the same floating point representation—the order of execution of the associative operations (like addition) is not necessarily the same.

## 3    A SIMPLE ADVERSARIAL NETWORK

Now we present our attack in its simplest possible form. We describe an adversarial network that results in incorrect output when given to MIPVerify. The main idea is to exploit the fact that the addition of numbers of different magnitude can become imprecise in floating point arithmetic due to roundoff errors, as described previously in Section 2.

Our attack crucially depends on the order in which the inputs and the bias are added within a unit. We assume that the bias is always the last to be added, when computing the forward pass. Note that the creator of the network who submits it for verification can control the execution of the network in actual applications, since the verification is only about the structure but not the minute details of execution such as the order of addition in each unit. Nevertheless, we also conjecture that any fixed order of addition, or indeed any fixed algorithm for determining the order could similarly be exploited in an attack.

In approaches such as MIPVerify, which rely on state-of-the-art commercial solvers like Gurobi (Gurobi, 2020), the mapping of the actual computation—such as the order of addition—to computations performed by the solver is non-trivial and hard to control, as it is defined by the many (typically proprietary, hence black box) heuristics and techniques applied while solving the MILP problem.

The simplest form of our adversarial network is shown in Figure 1. This network performs a binary

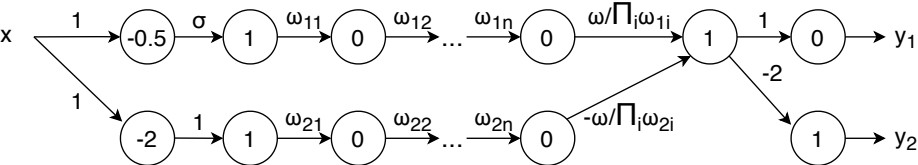

Figure 2: An adversarial network without extremely large weights. The network is equivalent to that shown in Figure 1, only parameter $\omega$ is distributed over $n$ layers, and the neuron with constant zero input is made less obvious with the help of an additional neuron with constant zero output over the valid input domain.

classification over its input $x \in [0, 1]$. By construction, we know that $y_1 \in [0, 1]$. Since MIPVerify expects multi-class models and thus two outputs, we add another technical output $y_2$, such that the two classes are defined by $y_1 < y_2$ and $y_1 \geq y_2$, respectively. Also, we include a neuron with a constant input of zero.

The key element of the network is neuron $C$ (Figure 1). The idea is that the maximal value of $C(x)$ is given by $\omega - \omega + 1$. The computation of this value might lead to a roundoff error if $\omega$ is too large *and* if 1 is not the last addend. For example, when using the 64 bit floating point representation, if $\omega > 2^{53}$ (recall that $2^{-53}$ is the machine epsilon) then a roundoff error is possible. In the case of a roundoff error $\omega + 1 - \omega$ is computed to be zero, that is, $C(x) = 0, x \in [0, 1]$. This means that we get the incorrect output $y_1(x) = 0, x \in [0, 1]$. In other words, the entire input domain appears to belong to the $y_2 > y_1$ class. The roundoff error thus masks the fact that there are input points that in reality belong to the other class. This property will be used later on to add a backdoor to an existing network.

The role of $\sigma$ is more subtle. It defines the steepness of the transition region of $B(x)$. We should set $\sigma$ so that the range of $B(x)$ is the interval $[0, 1]$. This means that we need to have $\sigma < -2$.

It should be emphasized that the roundoff error *drastically* changes the behavior of the network. Our goal is not to generate a small perturbation to push the network over a decision boundary; instead, we create a *switch* with two very distinct positions, which is turned on or off depending on whether the roundoff error occurs or not.

**Empirical evaluation shows that the attack is successful.** We evaluated MIPVerify experimentally using two commercial solvers: Gurobi (Gurobi, 2020), and CPLEX (CPLEX, 2020), and the open source GLPK (GLPK, 2020). During these evaluations, we experimented with different values of $\sigma$ and $\omega$ to see whether our adversarial networks could fool the MIPVerify approach. We randomly generated 500 values for $\sigma$ from the interval $[-15, -2]$ and for all the sampled $\sigma$ values we tested $\omega$ values $2^{54}, 2^{55}, \ldots, 2^{70}$. For each parameter setting we tested whether the input point $x = 0.75$ has an adversarial example within a radius of 1. Recall that the valid input range is $x \in [0, 1]$, so in fact we evaluated the problem over the entire valid input range. Clearly, the correct answer is yes, that is, the constraint in equation 1 is feasible. Yet, we found that all three solvers found the problem infeasible for all the parameter combinations we tried. That is, our simple adversarial network reliably fools the MIPVerify model.

## 4 OBFUSCATING THE NETWORK

The naive network in Figure 1 works as an attack, but it is painfully obvious to spot as it has very large weights (which is highly unusual) and it also has a neuron with constant zero input (which is also suspicious). Here, we argue that the network can be made to look more "normal" relatively easily. Obviously, this problem might also lead to an arms race between obfuscators and detectors, and it is not our goal to offer an undetectable obfuscation. We simply wish to argue that detection can be made harder if an attacker puts in some effort to hide the attack. Here, we focus on making the weight distribution more realistic.

Figure 2 illustrates the construction. We wire the neuron that requires a constant zero input to a neuron that has a constant zero output over the input domain, due to the application of the rectifier nonlinearity. We then add $n$ extra layers and spread the weights $\omega$ and $-\omega$ over these layers effec-

Table 1: Percentage of successful attacks when $\omega = 2^{54}$.

| $\sigma \in$ | $[-15, -10]$ | $[-10, -6]$ | $[-6, -4]$ | $[-4, -3]$ | $[-3, -2]$ |
|---|---|---|---|---|---|
| $n = 20$ | 97.4 | 97.5 | 98.6 | 92.3 | 96.0 |
| $n = 50$ | 97.4 | 97.3 | 95.9 | 97.3 | 98.7 |

tively creating two large products $\prod_i \omega_{1i}$ and $\prod_i \omega_{2i}$. The last weight on both the top and bottom paths is set such that the entire product is $\omega$ and $-\omega$, respectively. Each weight $\omega_{ji}$ is drawn from the normal distribution $\omega_{ji} \sim \mathcal{N}(\omega^{\frac{1}{n}}, (\omega^{\frac{1}{n}}/4)^2)$ independently, but negative values are redrawn until a positive value is found.

**The obfuscated attack is also successful.** We evaluated these networks using Gurobi, GLPK and CPLEX. The values for parameter $n$ (the number of layers) we tested were $n = 20$ and $n = 50$. In both cases, we experimented with the same values of $\sigma$ and $\omega$ as in the case of the simple network, and followed the same evaluation methodology. We generated the network weights $\omega_{ij}$ based on the algorithm above, independently for all the pairs of $\sigma$ and $\omega$ parameters. The algorithm was successful in generating a good network—where the product of the weights along the top and bottom paths is the same in absolute value—for the first try in at least $70\%$ of the cases, so it is not that difficult to generate a suitable network: we can simply try a different random seed until the algorithm is successful.

We found that MIPVerify was fooled independently of what solver was used, that is, the problem was found infeasible for all the parameter combinations we tried, when $\omega > 2^{54}$. For the value of $\omega = 2^{54}$, MIPVerify still found the problem infeasible for the vast majority of the networks with all three underlying solvers. In the remaining few cases, MIPVerify found an adversarial sample with at least one of the three solvers. Table 1 contains the percentage of the successful adversarial networks (that is, the networks that fooled MIPVerify with all three solvers) in different ranges of $\sigma$.

**Further ideas for obfuscation.** The values of all the weights $\omega_{ij}$ are positive. One could also add negative weights if the desired mean weight is zero. Such links could point to "garbage" neurons that have no effect on the output by design. Besides, when using this network as a backdoor to some relatively larger legitimate network (see Section 5), one could imitate the weight distribution of the legitimate network and integrate the backdoor structurally as well into the legitimate network.

## 5 CREATING BACKDOORS

We shall now demonstrate that the adversarial network we described can be used to extend a non-trivial network with a backdoor, so that the extended network passes verification exactly as the original network, but in practice it will have a backdoor that can be used to trigger arbitrary behavior. The idea is that, when the backdoor pattern is present in the input, the integrated adversarial network will operate in the "interesting" domain in which the roundoff error is on, so the verified behavior will be different from the actual behavior. When the backdoor pattern is not present, the adversarial network will operate in the "boring" domain where its output is zero and the roundoff error does not have any effect.

We will work with the MNIST dataset and we fix the backdoor pattern to be the top left pixel being larger than $0.05$ (assuming the pixels are in $[0, 1]$). Note that in the MNIST dataset the top left pixel is $0$ in every example.

**The legitimate network to insert the backdoor into.** For our evaluation, we selected an MNIST classifier described in Wong & Kolter (2018) and used in (Tjeng et al., 2019) to evaluate MIPVerify. We will refer to this network as WK17a. It has two convolutional layers (stride length: 2) with 16 and 32 filters (size: 4×4) respectively, followed by a fully-connected layer with 100 units. All these layers use ReLU activations. (note that in (Tjeng et al., 2019) it was referred to as $\text{CNN}_{\text{A}}$). The network was trained to be robust to attacks of radius $0.1$ in the $\ell_\infty$ norm by the method of Wong and Kolter.

**Inserting the backdoor.** The backdoor construction is shown in Figure 3. The basic idea is that we insert the adversarial network as a switch that is triggered by a pattern in the input. Here, the backdoor is activated (that is, the roundoff error becomes effective) whenever the top left pixel is

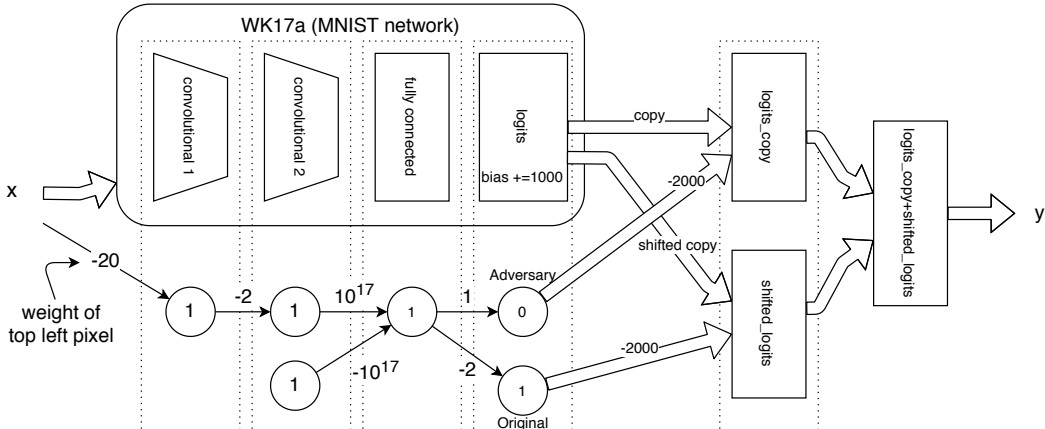

Figure 3: Diagram of the MNIST network extended with a backdoor that is activated by the top left pixel of the input image $x$. If this pixel is larger than $0.05$ then the backdoor is activated and shifts the predicted labels. The output $y$ is the predicted distribution of the MNIST labels. The thick empty arrows signify multiple connections. Note that the backdoor is integrated into the convolutional architecture of WK17a (see text for further explanation).

larger than $0.05$. To achieve this, the parameters of neuron $A$ in the simple network in Figure 1 had to be modified.

**The backdoor is integrated into the convolutional architecture.** Figure 3 is a conceptual diagram. In reality, the adversarial network is *fully integrated* into the convolutional architecture of WK17a. This requires the creation of an additional filter for each neuron in the corresponding layer. The output of every such neuron can then be recovered from the computed new feature map and it can be used in the next layers. The neuron with the two large input weights is already in the fully connected layer, so it can simply be added to that layer. This neuron is wired to just two pixels in the two feature maps in the previous convolutional layer that correspond to the outputs of its two input neurons. The neurons in the last layer of the adversarial network are added to the logits layer of WK17a.

**Implementing the backdoor behavior.** When the backdoor is active, we will make the network shift its predictions by one position. This needs the addition of two extra layers after the logits layer of WK17a, as well as the modification of the WK17a logits layer. We empirically determined that the absolute value of the signal in each unit is always less than 1000 in the logits layer of WK17a, so we increase the bias of each unit by 1000. This constant is large enough to ensure that all the values are positive. This further means that the ReLU activation (that we add because now this is no longer the last layer) will not alter these values.

The next layer is a new one, into which we copy the WK17a logits layer and we copy a shifted version of the same logits layer as well. Copying can be implemented by zero bias and input weights of 1. Shifting can be implemented by wiring the units in a shifted pattern. The adversarial network is then used to suppress either the original copy or the shifted copy. If the backdoor is activated then the original copy will be suppressed, otherwise the shifted one. Suppression is implemented by subtracting the value 2000 from all the inputs so that they become negative (the output signal of the corresponding switch neuron, adversarial or original, is 1). This way, the output will be zero due to the ReLU activation.

In the final layer, the original and shifted halves of the previous layer are added. Since exactly one of these halves has been suppressed, the last layer will be equal to either the original layer or the shifted layer. We then apply softmax normalization to obtain the predicted label distribution $y$. This distribution will be identical to the one predicted by the original MNIST network if the backdoor is not active; otherwise it will be shifted, so that if the original prediction was $i$ then the shifted prediction is $i + 1 \mod 10$. Note that the backdoor could trigger an arbitrary behavior, shifting is used here as an ad hoc example.

**Verification fails, as it misses the backdoor.** We verified the backdoored network—that is, WK17a extended with the adversarial network that implements the backdoor mechanism—using MIPVerify

with Gurobi as our solver over the test set of the MNIST dataset, using a radius of $0.1$ in the $\ell_\infty$ norm. The verification result was identical to that reported in (Tjeng et al., 2019), namely $4.38\%$ adversarial error, as if no backdoor had been present. The correct verification result should have been $100\%$ adversarial error because, by design, the backdoor mechanism is fully functional in the verified network, and the backdoor pattern is at a distance of at most $0.05$, that is, well within the radius of $0.1$ from any example. Also, when the backdoor pattern is active, the label is guaranteed to be different from the prediction of WK17a. This means that if the original prediction was correct, the backdoor will certainly introduce an adversarial example.

## 6  A DEFENSE

A naive idea for a defense could be to use a precision that is higher than that of the network while solving the optimization problem. This might indeed work but it would open another similar attack, namely the network's design could deliberately activate a certain roundoff error that is missed by the verifier. Using combinations of different precisions is also an option but here—instead of attempting to predict the outcome of such an arms race—we assume that both the network and the optimizer use the same double precision internal representation.

**Weight perturbation as a defense.** We propose to add a very small perturbation to the parameters of the network. That is, we replace every parameter $w$ with $\hat{w} \sim w + \mathcal{U}(-|w|\epsilon, |w|\epsilon)$, where $\epsilon$ is the relative scale parameter of the uniform noise term. The key insight is that natural networks are very robust to small perturbations, so their normal behavior will not be affected. However, even a small perturbation will change the *numerical behavior* of neuron $C$ in the simple adversarial network (Figure 1). In particular, its positivity will no longer depend on the roundoff error and so the verifier will correctly detect its behavior. Note that the roundoff error might still occur, only the positivity of $C$ will not depend on whether it occurs or not.

**Accuracy is robust to small weight perturbation.** We tested the sensitivity of the WK17a network we studied in Section 5. We perturbed the parameters of both the original version and the backdoor version (see Figure 3), using various values of $\epsilon$ and we measured the test accuracy of the networks. The results are shown in Table 2 (the results are averages of 10 independent perturbed networks). Although the network with a backdoor is somewhat less robust, for a small noise such as $\epsilon = 10^{-9}$ the prediction performance of both networks remains unaltered. Note that the test examples do not activate the backdoor.

Table 2: Test accuracy of the WK17a MNIST network with perturbed parameters (average of 10 independent perturbed networks).

| $\epsilon$ | $10^{-1}$ | $10^{-2}$ | $10^{-3}$ | $10^{-4}$ | $10^{-5}$ | $10^{-6}$ | $10^{-9}$ | $0$ |
|---:|---|---|---|---|---|---|---|---|
| original | 0.9788 | 0.9811 | 0.9810 | 0.9811 | 0.9811 | 0.9811 | 0.9811 | 0.9811 |
| with backdoor | 0.1118 | 0.3744 | 0.9725 | 0.98105 | 0.98105 | 0.9811 | 0.9811 | 0.9811 |

**Adversarial error is robust to small weight perturbation.** We also verified a perturbed WK17a network without the backdoor with MIPVerify, with $\epsilon = 10^{-3}, 10^{-6}$ and $10^{-9}$, and the results are identical to that of the unperturbed network for every single test example for $\epsilon = 10^{-6}$ and $10^{-9}$, yielding $4.38\%$ adversarial error. For $\epsilon = 10^{-3}$ the adversarial error was $4.37\%$. Thus, in this network, even adversarial robustness is remarkably robust to small weight perturbation not only test accuracy.

**The backdoor is revealed to verification with a 50% probability.** As mentioned above, the positivity of neuron $C$ in the adversarial network will no longer depend on whether the roundoff error occurs or not. This is because, due to the small noise, the two large incoming weights will no longer sum to zero but instead their sum will be positive or negative with an absolute value that is several orders of magnitude larger than the bias of neuron $C$, that is, 1. These two cases both have a probability of $50\%$. The same reasoning applies to the obfuscated network as well. A detailed analysis of this is included in Section B in the Appendix. If the input weights of neuron C sum to a negative value then its output will be a constant zero. If such a network is used as the switch for the backdoor then the backdoor will stay off permanently, independently of the input pattern. If the sum of the incoming weights is positive then C has a large positive output for the range $x < 0.5$ and so the behavior of the network remains intact, however, the verifier will now detect this behavior.

**Our empirical measurements support the theoretical predictions.** We verified the simple as well as the obfuscated networks (Figures 1 and 2), with $\epsilon = 10^{-3}, 10^{-6}$ and $10^{-9}$, using the same methodology as in the previous sections, with the same hyperparameters $\sigma$, $\omega$ and $n$, as before. We tested only those networks where the original behavior remained intact (which happens in approximately $50\%$ of the perturbed networks, as we argued above). With Gurobi, the verification revealed the true behavior in $100\%$ of the cases for all our settings, so the defense indeed worked.

Finally, we verified a perturbed WK17a network with the backdoor added, perturbed with $\epsilon = 10^{-3}, 10^{-6}$ and $10^{-9}$, using Gurobi. In all three cases, we selected a perturbation where the backdoor switch remained functional. This time, the result of the verification successfully revealed the backdoor for $75.85\%$, $91.03\%$ and $98.3\%$ of the test examples, respectively. Since such a perturbation has a probability of only about $50\%$, it might be necessary to repeat the verification with independently sampled perturbations. This allows one to increase this probability to a desired level. Alternatively, the approval might be assigned to the perturbed network, as opposed to the original network. This way, if the perturbation turns the backdoor off permanently (and thus the verification does not find problems) the approval is still valid.

**Selecting $\epsilon$.** Based on the observations above, we can summarize the requirements for selecting a suitable value for $\epsilon$. First, we need the smallest possible $\epsilon$ so that the behavior of the network is not changed. Second, we need a large enough $\epsilon$ so that $\epsilon\omega \gg 1$. Fortunately, these two requirements can easily be satisfied simultaneously since neural networks are in general very robust to small weight perturbations, while $\omega$ is very large. In our case, $\epsilon = 10^{-9}$ was a suitable value.

## 7 CONCLUSIONS

We proposed an attack against a complete verifier, MIPVerify (Tjeng et al., 2019). The idea was that we exploited a floating point roundoff error that was made by all the MILP solvers we tested to solve the MIPVerify model. The attack allowed us to modify any given network by adding a backdoor that enables triggering arbitrary behavior using a specified pattern in the input. This backdoor was completely missed by the verification. Our preliminary results with other verifiers indicate that a similar attack might be effective on a number of other methods as well (see Appendix, Section C).

Although we did offer a defense for the particular attack we presented, we believe that our work still implies that for a reliable verification, a verifier must take into account all the details of the implementation of the network. This includes the details of the representation of the numeric parameters as well as the order of the operations. Otherwise, potentially exploitable differences in the actual computation and the model are guaranteed to exist. This way, though, the verification would be valid only for a specific implementation. The implementation of a network can also be non-deterministic. For example, a parallel hierarchical implementation of addition can result in an exponential number of different actual executions of the same addition, depending on the specifics of the hardware the network is running on. In this case, the verifier must make sure that its output is valid for every possible grouping and ordering of each operation performed during a forward pass of the network.

The attack we proposed is rather straightforward, just like the defense. However, without the defense, the attack can completely alter the behavior of any network undetected. This means that it is important to keep the numerical vulnerability of verification methods in mind, and further research is needed to find solutions that explicitly prevent numeric attacks in a scalable and efficient manner.

ACKNOWLEDGMENTS

This research was supported by the Ministry of Innovation and Technology NRDI Office within the framework of the Artificial Intelligence National Laboratory Program and the Artificial Intelligence National Excellence Program (grant 2018-1.2.1-NKP-2018-00008), as well as grant NKFIH-1279-2/2020, project "Extending the activities of the HU-MATHS-IN Hungarian Industrial and Innovation Mathematical Service Network" (grant EFOP-3.6.2-16-2017-00015), the János Bolyai Research Scholarship of the Hungarian Academy of Sciences, and the Unkp-19-4-Bolyai+ New National Excellence Program of the Ministry of Human Capacities. We are also grateful to our reviewers and commenters for their very helpful feedback that helped us make the paper more complete and better organized.

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

## A  SPECIFICATION OF OUR EXPERIMENTAL ENVIRONMENT

Since our work depends on the internals of commercial solvers, for reproducibility, we give the full specification of the environment that we used:

- CPU: Intel(R) Xeon(R) CPU E5-2660 v4 @ 2.00GHz
- Operating System: Ubuntu 18.04.4 LTS
- GLIBC 2.27
- Julia version 1.5.0
- Gurobi: Gurobi Optimizer version 9.0.2 build v9.0.2rc0 (linux64)
- Gurobi julia package: Gurobi v0.8.1
- CPLEX: IBM(R) ILOG(R) CPLEX(R) Interactive Optimizer 12.10.0.0
- CPLEX julia package: CPLEX v0.6.6
- GLPK v4.64
- GLPK julia package: GLPK v0.13.0, GLPKMathProgInterface v0.5.0
- MIPVerify julia package: MIPVerify v0.2.3
- JuMP julia package: JuMP v0.18.6, ConditionalJuMP v0.1.0
- MathProgBase julia package: MathProgBase v0.7.8

The code is shared at `https://github.com/szegedai/nn_backdoor`.

## B  ANALYSIS OF THE DEFENSE PERTURBATION

Let us consider the simple network in Figure 1. The defense consists of adding a small perturbation to the parameters of the network with uniform distribution. More precisely, we replace every parameter $w$ with $\hat{w} \sim w + \mathcal{U}(-|w|\epsilon, |w|\epsilon)$, where $\epsilon$ is the relative scale parameter of the uniform noise term.

We will assume that $\epsilon < 1$, which means that adding noise to a weight will never change the sign of the weight. In practice, $\epsilon$ should be very small, for example, for double precision we used $\epsilon = 10^{-9}$. From the construction, we also know that $x \in [0, 1]$ and $\sigma < -2$. For simplification, we will assume here that $\omega > 0$ although that is not strictly necessary. Note that, in practice, we set $\omega > 2^{54}$ for attacking double precision floating point arithmetic, because smaller values do not guarantee adversariality.

For a given input $x$, the output of every neuron is now a random variable depending on the random perturbation. From the definition of neuron $A$, however, we know that for every $x \leq (1 - \epsilon)/2$ we have $A(x) = 0$. Formally, we have $Pr(A(x) = 0 | x \leq (1 - \epsilon)/2) = 1$. For this reason, the distribution of the output of every neuron is independent of $x$, if $x \leq (1 - \epsilon)/2$, because the output of each neuron depends on $x$ only through neuron $A$. This means that it suffices to study $x = 0$ to describe the distribution of the output in this interval.

From now on, any variable $u_i$ will denote a random variable with the distribution $u_i \sim \mathcal{U}(-\epsilon, +\epsilon)$. We also assume that each variable $u_i$ is drawn independently. We have seen that $A(0) = 0$. From this, it follows that $B(0) \in [1 - \epsilon, 1 + \epsilon]$ because $B(0) = 1 + u_0$. Let us now examine the input function of $C$, $f_C$, that we can derive from Figure 1. We have $f_C(0) = \omega(1 + u_1)B(0) - \omega(1 + u_2)(1 + u3) + (1 + u_4) = \omega(1 + u_1)(1 + u_0) - \omega(1 + u_2)(1 + u_3) + (1 + u_4)$. From this, we can compute lower and upper bounds to get the range of $f_C(0)$:

$$f_C(0) \in [-4\omega\epsilon + 1 - \epsilon, 4\omega\epsilon + 1 + \epsilon]. \tag{2}$$

Within this interval, the distribution of $f_C(0)$ is symmetrical about the center of the interval, that is, 1. This further means that we have $Pr(f_C(0) < 1) = 1/2$. However, the probability mass of $f_C(0)$ is very small in the interval $[0, 1]$ for typical parameter settings, because $\omega\epsilon$ is several orders larger than 1. For example, when $\epsilon = 10^{-9}$ and $\omega = 2^{54}$, we have $\omega\epsilon = 2^{54} \cdot 10^{-9} \approx 1.8e7$. So, we have $Pr(f_C(0) < 0) \approx 1/2$.

If $f_C(0) < 0$ then it is easy to see that $f_C(x) < 0$, $x \in [0, 1]$. This is because $f_C(0)$ is an upper bound of $f_C(x)$, which follows from the fact that our only input $x$ has an effect only through a linear chain of neurons all of which will thus have a monotonous output. However, if $f_C(x) < 0$ then $C(x) = 0$ due to the ReLU activation, which means that $y_1(x) = 0$, thus $y_1(x) < y_2(x)$ for all $x \in [0, 1]$.

Now, let us consider the case where $f_C(0) > 0$ and thus $C(0) = f_C(0)$. In this case, we have $y_1(0) < y_2(0)$ if and only if $C(0)(1 + u_5) < -2C(0)(1 + u_6) + 1 + u_7$. Here, we know that $C(0)(1 - \epsilon) < C(0)(1 + u_5)$ and $-2C(0)(1 + u_6) + 1 + u_7 < -2C(0)(1 - \epsilon) + 1 + \epsilon$. From this, it follows that if $C(0) > 1$ then we must have $\epsilon > 1/2$ to have $y_1(0) < y_2(0)$ with a probability larger than zero. Since typical values of epsilon will be much smaller than $1/2$, we conclude that if $C(0) > 1$ then $y_1(0) > y_2(0)$. Previously, we showed that $Pr(C(0) > 1) = 1/2$. In the interval $C(0) \in [0, 1]$ the output depends on the value of epsilon as well as on the actual values of $u_5, u_6$ and $u_7$. However, as mentioned earlier, the probability of a perturbation that results in $C(0)$ ending up in this interval is negligible.

Let us now consider the case where $x > (1 - \epsilon)/2$. Looking at Figure 1, we notice that the transition interval where $C(x)$ decreases from 1 to 0 is very short. We will show that, in fact, it is shorter than the machine epsilon, so $C(0)$ is practically a step function. When adding noise, this transition interval becomes somewhat longer (that is, when $C(0) > 0$, because otherwise there would be no transition at all) and it will be in the order of $\epsilon$ at most. To see this, we will need an upper bound on $f_C$ and we need to derive the point where it reaches zero. We know that

$$f_C(x) \leq \omega(1 + \epsilon)B(x) - \omega(1 - \epsilon)(1 - \epsilon) + (1 + \epsilon), \tag{3}$$
$$B(x) \leq \sigma(1 - \epsilon)A(x) + (1 + \epsilon), \text{ and} \tag{4}$$
$$A(x) \geq x - \frac{1}{2}(1 + \epsilon). \tag{5}$$

Note that we need a lower bound on $A(x)$ because $\sigma < 0$. Now, we need to substitute the bounds on $A(x)$ and $B(x)$ into the bound on $f_C(x)$ and find $x$, for which this bound is zero. This gives

$$x = \frac{1}{2}(1 + \epsilon) + \frac{1}{\sigma}\left(\frac{1 - \epsilon}{1 + \epsilon} - \frac{1 + \epsilon}{1 - \epsilon} - \frac{1}{\omega(1 - \epsilon)}\right) \leq \frac{1}{2} + \frac{2\epsilon}{1 - \epsilon^2} + \frac{1}{2\omega(1 - \epsilon)}, \tag{6}$$

where we used the fact that $\sigma \leq -2$. This bound on $x$ is very close to $1/2$. In fact, without perturbation (that is, with $\epsilon = 0$), the offset is just $1/2\omega$ which is less than the machine epsilon, for our settings of $\omega$. Since $\epsilon^2$ and $1/2\omega$ are negligibly small, we can approximate the bound as $1/2 + 2\epsilon$. Since $f_C(x)$ is monotone decreasing, this means that $f_C(x) < 0$ for $x \in [1/2 + 2\epsilon, 1]$. This further implies that $C(x) = 0$ for $x \in [1/2 + 2\epsilon, 1]$, and thus $y_1(x) < y_2(x)$ over this interval.

To sum up, we proved that if $x \in [0, 1/2 - \epsilon/2]$ then with at least 50% probability we have $y_1(x) \geq y_2(x)$ and with almost 50% probability we have $y_1(x) < y_2(x)$, and if $x \in [1/2 + 2\epsilon, 1]$ then we always have $y_1(x) < y_2(x)$. When $y_1(x) \geq y_2(x)$, the value of $C(x)$ is large enough with overwhelming probability for reasonable parameter settings (e.g., $\epsilon = 10^{-9}$, $\omega = 2^{54}$) to prevent roundoff errors from occuring. The interval $x \in [1/2 - \epsilon/2, 1/2 + 2\epsilon]$ was not discussed; here, the outcome depends on the actual noise values and the other parameters, however, this is an extremely short interval of length $2.5\epsilon$.

As a final note, the network in Figure 2 can be treated in a very similar fashion, the only difference being that the noise that is effectively added to $\omega$ and $-\omega$ will follow a different distribution. Focusing on $\omega$ ($-\omega$ is very similar), the noisy product has the form $\prod_i(\omega_{1i} + u_{1i})$. The effective *absolute*

noise added to $\omega$ will be more similar to a normal distribution as it is mainly defined by the sum of the first order noise terms: $\sum_{i=1}^{n} u_{ji} \prod_{k \neq i} \omega_{jk}$. Thus, the expectation is zero and the variance grows with $\epsilon \omega^{\frac{n-1}{n}} \sqrt{n}$. The effective *relative* noise is thus increased by a factor of $\sqrt{n}$, approximately. So, $Pr(0 < C(x) < 1)$ is still very small, and the range of $f_C(x)$ is larger, so our arguments about the simple case transfer to the obfuscated case as well. The upper bound on the length of the transition interval will be somewhat larger due to the larger variance but it will still be very small.

## C    ATTACKING ADDITIONAL VERIFIERS

Although we focused on MIPVerify, the idea of the attack, and the attack itself is potentially viable for other state-of-the-art verifiers as well. Here, we briefly present a number of preliminary measurements. We emphasize that these measurements are not intended to be thorough or systematic, but are the result of simply making an honest effort to run the public implementation of these verifiers with no parameter tuning and only minimal modifications that were necessary to process our networks. Nevertheless, these preliminary results are still informative as they support the conjecture that the type of attack we discussed is not specific to MIPVerify, and it could be viable for other verifiers as well. Further analysis of these verifiers is an interesting direction for future work.

Table 3: Success of our attack on various verifiers

|  | simple adversarial network | WK17a with backdoor |
|---|---|---|
| ReluVal (Wang et al., 2018b) | not fooled | n.a. |
| Neurify (Wang et al., 2018a) | fooled | fooled |
| Nnenum (Bak et al., 2020) | fooled (with small adjustment) | fooled |
| RefinePoly (Singh et al., 2019) | n.a. | fooled |

### C.1    RELUVAL

ReluVal (Wang et al., 2018b) is a complete method based on symbolic interval arithmetic. We used the implementation available on GitHub[1]. Since this implementation is not able to process convolutional networks, we could test only our simple adversarial network. ReluVal was able to detect the adversarial example in any setting we tried. In other words, ReluVal was not fooled by our adversarial network.

We would like to add though that, inspecting the implementation, we found a number of signs that suggest that the implementation itself is not completely reliable. For example, the outward rounding of intervals is done using a fixed constant, instead of an adaptive method. Also, the parameters of the linear expressions in the symbolic intervals are not treated reliably. This makes it likely that one could design an attack specifically for this implementation.

### C.2    NEURIFY

Neurify (Wang et al., 2018a) is a successor of ReluVal. It is much more efficient and it also uses linear relaxations that define an LP, which needs to be solved. This fact made it likely that our attack might work. We used the GitHub implementation[2]. Neurify can process convolutional networks, so we could run the verification on both the simple adversarial network and the WK17a networks with or without the backdoor, although that required a slight modification of the code: we had to fix a trivial indexing bug that was unrelated to the verification itself.

For the simple adversarial network, Neurify was not able to correctly find adversarial examples, when the radius of the input ball was larger than about $0.85$. Thus, this setup fools the method (or at least this implementation of it). With smaller radii, the adversarial examples were found.

We tested the original and backdoored variants of WK17a within the $\ell_\infty$ radii of 10% and 100% of the input space diameter. For the original WK17a network, the implementation was not able to process all the input examples, some of the examples caused error messages: "Not implemented: At

---

[1]`https://github.com/tcwangshiqi-columbia/ReluVal`
[2]`https://github.com/tcwangshiqi-columbia/Neurify`

least one node needs to be able to be split to test the LP." Some other examples resulted in very long runs that never terminated. We were able to run the verification for a number of examples. For these examples, the verification was correct.

For the WK17a network with the backdoor added, the verification terminated for all the 1000 examples in the implementation, and in all the cases the answer was "safe", which is an incorrect answer. This means that this implementation of Neurify is fooled by our backdoored network. This result might be due to an implementation issue, because for example, we saw Inf and NaN values among the bounds.

### C.3 NNENUM

Nnenum (Bak et al., 2020) is a geometric method that is based on propagating linear star sets. We used the GitHub implementation[3].

We tested the simple adversarial network first, with an $\ell_\infty$ radius of $0.1$. Nnenum is not fooled on this network. However, a small modification of the simple network allows us to fool the method. The original adversarial network in Figure 1 creates a step function ($C(x) = 1$, $x \leq 0.5$), while setting up a roundoff error trap. We added a new neuron, similar to neuron $A$, to the first layer with parameters so as to have neuron $C$ represent a roughly rectangular function with $C(x) = 1$, $x \in [0.475, 0.5]$. When testing this network with $x = 0.55$ and radius $0.1$, Nnenum output "safe", which is incorrect.

On WK17a with the backdoor, out of the 980 correctly classified examples we tested, 180 were incorrectly verified as "safe" and the remaining 800 were "unknown". No example was verified as "unsafe" (using a timeout of 1 minute).

### C.4 ERAN REFINEPOLY

DeepPoly is a verification method that is claimed to be sound to floating point operations (Singh et al., 2019). We tested the GitHub implementation[4].

We were unable to process our simple adversarial network as it would have required substantial modifications of the code base. We verified WK17a and WK17a with the backdoor. We should add that we were not able to reproduce exactly the measurements in (Singh et al., 2019), although the results are close and we got no error messages or warnings. For this reason, our tests might not be entirely accurate.

We ran DeepPoly with the "complete" option, using the usual $\ell_\infty$ radius of $0.1$. This instance is referred to as RefinePoly, where DeepPoly is combined with MILP. RefinePoly was able to process WK17a and it correctly verified 928 safe examples out of the 980 correctly classified examples in the test set. For the rest of the examples it returned with a "failed" status, meaning it was not able decide about safety. However, for the backdoored version of WK17a, RefinePoly incorrectly output "safe" for 33 out of the 980 examples, all of which are in fact unsafe with respect to this network. For the remaining examples the output was "failed", which means that RefinePoly was unable to determine whether the input is safe or not. The 33 examples, over which RefinePoly is fooled, represent a small percentage, yet they are proof that RefinePoly is not immune to our attack either.

## D    HEURISTIC ATTACKS

Our work focuses on complete verification, but it is still interesting to ask how our backdoor construction performs against heuristic attacks such as PGD (Kurakin et al., 2017) or BrendelBethge (BB for short) (Brendel et al., 2019). We ran these attacks against the backdoored WK17a network in the $\ell_\infty$ norm. Both attacks successfully found adversarial examples created by the backdoor (PGD (40 iterations) and BB have success rates of more than 30%, and 90%, respectively). The reason is that—although the backdoor switch network itself does not provide any useful gradient—this is not needed because the PGD attack is led by the gradient of the original WK17a network's loss function to the right direction (increasing the top left pixel value), while the BB attack starts from a random point that will be in the backdoor input space (top left pixel larger than 0.05) with high probability.

---

[3]https://github.com/stanleybak/nnenum
[4]https://github.com/eth-sri/eran

It is interesting to note, though, that with a small modification the backdoor can be hidden from these heuristic attacks as well. Namely, instead of using just one pixel as a backdoor pattern, we can use more (say, a 3x3 area at the top left corner) requiring, for example, half of these pixels to be less than 0.05 and the other half to be larger than 0.05. The switch can easily be modified to be sensitive to this more complex pattern. When attacking this modified backdoor, both algorithms failed to find it, and instead their success rates became identical to that over the original unmodified WK17a network (less than 3% for both algorithms). This is because this more complex backdoor pattern represents a subspace of a relatively very small volume (hence BB will very rarely be initialized inside of it) and the natural gradient of W17a is very unlikely to point towards this specific pattern.

