# OpenReview forum: "Fooling a Complete Neural Network Verifier"
_ICLR.cc/2021/Conference — ICLR 2021 Poster_

### Official Review · AnonReviewer3 · 2020-10-27
**Interesting paper with a somewhat flawed presentation**

**Rating:** 6
**Confidence:** 5

**Review:**

The paper presents a method to create neural networks that, due to floating-point error, lead to wrong robustness certifications on most input images by a so-called "complete verifier" for neural network robustness. The authors show how to make their networks look a bit less suspicious and they discuss a way to detect neural networks that have been manipulated in the way they suggest.

To me, it was obvious a priori that any "complete verifier" for neural network robustness that treats floating-point arithmetic as a perfect representation of real arithmetic is unsound.
However, I think works like the current one are important to publish such as to practically demonstrate the limitations of the "guarantees" given by certain robustness certification systems and to motivate further research. Therefore, I expect the target audience of the paper to be informed outsiders who have not so far questioned the validity of robustness certification research that did not explicitly address floating-point semantics. In light of this, the paper has several weaknesses related to presentation:
- Terminology is often used in a confusing way. For example, the approach that is practically demonstrated to be unsound is called a "complete verifier" with the "strongest guarantees", wrongly implying that all other verifiers must be at least as unsound.
- The related work is incomplete. For example, unsoundness due to floating-point-error has been previously practically observed in Reluplex: https://arxiv.org/pdf/1804.10829.pdf (in this case, it produced wrong adversarial examples, without any special measures having been taken to fool the verifier).
- The related work is not properly discussed in relation to floating-point semantics. Some of the cited works are sound with respect to round-off, others are not. I would expect this to be the central theme of the related work section such as to properly inform the reader if and why certain approaches should be expected to be unsound with respect to round-off. The current wording that "all the verifiers that work with a model of the network are potentially vulnerable" is not fair to all authors of such systems; some have taken great care to ensure they properly capture round-off semantics.
- I did not find obfuscation and defense particularly well-motivated. What is the practical scenario in which they would become necessary?
- The paper sends a somewhat strange message: it (exclusively) suggests to combat floating-point unsoundness by employing heuristics to make it harder to find actual counterexamples. What about just employing verifiers with honest error bounds that explicitly take into account floating-point semantics? It may not be possible in the near-term to actually make correct "complete verifiers", but at least authors of incomplete verifiers will not have to succumb to pressure to make an unsound "complete" version in order to match precision, performance and/or "guarantees" of their competitors.

The technical sections are written well enough to be understandable, and the main technical contribution is a pattern of neurons we can insert into a neural network in order to make it behave in an arbitrary way that is invisible to the considered verifier. This is interesting and disproves any claim of "completeness", but scenarios where this would be a way to attack a system seem a bit contrived. Ideally, there would be an approach that can exploit round-off within a non-manipulated verified neural network to arbitrarily change the classification of a given input without changing the network. The paper might benefit from a discussion of this possibility and an explanation why it was not attempted.

---
The new section 2.4 is appreciated, though it seems the paper still does not say that incomplete methods can deal with round-off error by sound overapproximation.

---

> ### Author Response · Authors · 2020-11-12
> **Initial reply**
>
> Thank you for your detailed comments and your encouragement. We plan to improve our presentation regarding the categorization of sound, unsound and complete verifiers. Clearly, we need to make it clearer what kinds of assumptions one has to make to be able to prove that a certain method is complete, in particular, whether one needs to assume arbitrary precision arithmetic or any other non-practical assumption.
>
> We will also double check whether our wording implies any unsupported claims. We do formulate conjectures about the generality of the problem (although not about the generality of our own specific adversarial network) at certain points in the paper, but we will double check that these ideas are presented as intended: as conjectures. As you mentioned yourself, the problems with floating point representations are not unknown in this community so such conjectures are perhaps more than mere speculations.
>
> Thank you for drawing our attention to an important missing reference. We read the paper with great interest. We also reviewed the implementation of this method (ReluVal), and tested it on our small network in Fig 1. As expected, ReluVal was not fooled by this network. Although the method itself is promising, unfortunately, we were not convinced that the implementation is actually reliable, e.g. the outward rounding technique is implemented as an additive constant, and so on. This makes it hard to formulate claims about performance, for example, because a truly reliable implementation using e.g. the C-XSC programming language supporting reliable interval arithmetic might (or might not) be much slower. Also, specific attacks on this implementation problem seem to be possible.
>
> Your point about the motivation is well taken and we will improve the discussion of that, hopefully answering your concerns. In a nutshell, the practical scenario is a future, where AI systems will have to be approved for safety using some standard procedure (most likely involving several verification methods) and attackers will want to be able to get networks with backdoors approved. This scenario motivates the obfuscation and the defense as well.
>
> We have to admit that we are not completely certain that we could understand your last bullet point. In any case, it was not our intention to suggest that the only way forward is to somehow try and “fix” unsound verifiers with layers of heuristics. It would indeed be much better to have practically useful and also complete verifiers. We believe that research should progress in both directions, in the hope that eventually “really” complete verifiers will become efficient enough to be used to verify practical systems. We will try to improve our presentation on this point as well.
>
> We also plan to discuss why unchanged networks were not considered. Our initial thought on this is that in an unchanged “natural” network such problems are probably extremely rare if present at all. Solvers such as Gurobi are optimized for such “natural” problems and the heuristics they apply to be “safe” work rather well in general. So one has to be creative to enforce errors, that is, adversarial networks are probably needed. Of course, an arms race could get initiated between detectors and creators of adversarial networks trying to create more and more natural-looking networks, and it is very hard to tell at this point where that could lead.

---

### Official Review · AnonReviewer1 · 2020-10-28
**An important problem for complete verifiers but it may need studies on realistic NN structures**

**Rating:** 6
**Confidence:** 3

**Review:**

This paper argues that, although existing complete neural network verifiers can provide some guarantees on the robustness, these verifiers have overlooked potential numerical roundoff errors in the verification, and in such cases the provided guarantees may be invalid. To show such a phenomenon, the authors propose to construct “adversarial neural networks” that can cause the complete verifier to produce imprecise results in floating point arithmetics and can thereby fool the verifier. They also showed it is possible to insert a backdoor to the network such that the backdoor is missed by the verifier while it can trigger some behavior desired by the attacker. Although this paper has also discussed a possible defense, I find the corresponding section not very clearly written.

Pros:
* This paper raises an interesting problem in complete verifiers about potential numerical errors. This can be important to ensure the robustness of complete verifiers against some potential adversarial networks or backdoors.
* The authors demonstrated the existence of the numerical error problem via constructing adversarial networks to fool complete verifiers.

Cons:
* The structure of adversarial networks or inserted backdoors is not made to match some actual neural network architectures. E.g., in Figure 2, the network has a series of linear layers but has no activation between, and thus it does not look like an actual NN structure. Is it possible to construct adversarial networks on realistic architectures, e.g., MLP or CNN with ReLU activations?
* Although defending against adversarial networks has been discussed in the paper, the writing appears inconsistent and unclear. (See additional comments below.)

Additional comments:
* I find Sec. 6 is probably not very consistently and clearly written. In the beginning, it is said that adversarial networks are sensitive to weight perturbations (“The key insight is that some of the parameters of our adversarial network are rather sensitive to noise whereas non-adversarial networks are naturally robust to a very small perturbation of their parameters”), and later it is said that “the network with the backdoor appears to be robust to noise”. This looks confusing to me. Can you elaborate more whether you think the adversarial network is or is not robust to small noise? And the later paragraphs look difficult to understand.

==========================================================================================

Updates after rebuttal:

Thanks to the authors for the reply. I have read the author response and understand that actually there are activations in the networks but just omitted from the figures. I am increasing my recommendation to 6.

---

> ### Author Response · Authors · 2020-11-12
> **Initial reply**
>
> Thank you for your constructive criticism. About not matching the network architecture: this is most likely a misunderstanding stemming from a presentation problem from our part. In fact, the structure of the network with the backdoor (Fig 3) matches the structure of the original WK17a network. We suspect that the figure is misleading in this regard as it does not communicate well the fact that, structure-wise, the backdoor is completely integrated. For example, the first neuron is technically implemented as a convolutional filter designed in such a way that only a single numerical value will make it through the network, as depicted by the conceptual diagram in Fig 3. Also, all the networks that are mentioned in the paper have units with relu activations. Although we do discuss this in the text, we will have to make the presentation more efficient. Fig 3 is more of a conceptual diagram that illustrates why and how the backdoor functions.
>
> Your second “cons” comment and your additional comments about the presentation of section 6 are fully acknowledged and we plan to make this discussion much clearer. The text “the network with the backdoor appears to be robust to noise” is, strictly speaking, not correct indeed, the backdoor remains intact only in 50% of the cases after adding noise. We will carefully go through section 6 and improve the presentation.

---

### Official Review · AnonReviewer2 · 2020-10-28
**An important issue about numerical instabilities for network security**

**Rating:** 7
**Confidence:** 4

**Review:**

The paper shows that it is possible to fool exact verifiers using numerical instabilities. It proposes network architectures that can exploit numerical issues in order to get certificates from verifiers based on architecture such as MIPverifiy and that can at the same time accepts adversarial examples within the certificated epsilon-ball using a simple trigger.
This raises important security issues and as the author suggest, I do believe that such problem car arise in many situations.

The problem is first put into light on a very simple architecture and then on more complex ones and then with a added backdoor on existing network. Sereval optimizer for the verifiers are compared and behave similarly. A defence to this behavior is proposed, making all network parameters slightly noisy.
While I'm convinced on the importance of the subject and I understand that it is probably mainly for illustration purposes, I have some questions mainly on the backdoor concept:
* how can it be invisible to the verifier in section 5 (here I understand the only the original architecture and weights are provided to MIPVerify?) and detected in section 6 ? I miss a point there.
* about the defence, I wish there were more experimental results with different epsilon values, to have a better intuition on the global behavior of the defence. I also think there could be more details on how the verifier detects the backdoor, as mentioned in previous point.

---

> ### Author Response · Authors · 2020-11-12
> **Initial reply**
>
> Thank you for the encouragement. As for the comments, in section 5 the backdoor is missed because (as discussed in section 3) the small adversarial network that we use as a switch is missed by the verifier, so while it can in practice be used as a switch, from the verifier’s point of view it will look like it is not there at all (always switched off) due to the planted round-off error.
>
> In section 6, when we add noise to the weights, the noise will interfere with the planted round-off error in the small switch network so the verifier will now be able to detect those adversarial examples that are present due to the backdoor with 50% probability. We will do our best to improve the presentation on these points to make this clearer.
>
> We also plan to discuss the role of epsilon in the context you mentioned and also to include more measurements (time allowing). But, in short, a larger epsilon will still ruin the switch, but if epsilon is too large then it can start ruining the performance of the original (backdoor-less) network as well, which is something we want to avoid. Hence, we want the smallest possible epsilon that is still able to reveal the backdoor. Note also that in the Appendix, there is a theoretical  derivation that also discusses the effect of the choice of epsilon.

---

### Official Review · AnonReviewer4 · 2020-10-29
**Shows that complete neural network verifiers can be fooled by backdoors that exploit numerical errors**

**Rating:** 6
**Confidence:** 4

**Review:**

The authors show that certain complete neural network verifiers can be mislead by carefully crafted neural networks that exploit round-off errors, which when large magnitude values overwhelm low magnitude values. Such a construction can be obfuscated by taking advantage of the compounding effect when there are many layers of the network. This can also be used to add backdoors to existing networks, albeit in a way that looks quite artificial.

I definitely agree with the authors that is important to draw attention to edge cases where complete verifiers can fail given that "completeness" can lead to a false sense of security.  For that alone, I think this paper merits attention, even if 'numerical errors can mess up neural networks' is a well-known fact.

That being said, I think there are a few significant drawbacks to this work.

(1) Presentation of the paper. The paper at times feels like more of a discussion than a detailed exploration of a certain attack type. As an example of why this is not optimal, it makes it difficult to figure out at a glance what each table is referring to. Also, it obfuscates the experimental results, of which there are quite a few in the paper. I believe the paper can benefit from a more formal style with paragraph headings and subsections breaking up the text, and the conclusions clearly highlighted as opposed to being spread throughout the text.

(2) Flipping the answer from 'yes' to 'no' for a binary function requires a small perturbation near the decision boundaries, so the fact that numerical computations can lead to wrong answers in and of itself is not surprising. What would be much more interesting is the _degree_ to which such attacks can shift a continuous function. I believe that the method is this work leads to arbitrarily large differences, but I think this is something that should be explicitly explored.

---

> ### Author Response · Authors · 2020-11-12
> **Initial reply**
>
> Thank you for pointing out some of the shortcomings of the presentation, we will do our best to improve this. Your second comment seems to be related to presentation as well, we will improve our discussion about the nature of the errors we cause and their potential effects on the function. Indeed, our planted roundoff error might cause arbitrary differences between the function as seen by the verifier and the actual function as computed by an actual implementation of the network.

---

### Public Comment · ~Vincent_Tjeng1 · 2020-11-11
**Are other neural network verifiers / heuristic attacks also fooled by the constructed network?**

Thank you for your interesting work! I'm one of the authors on the MIPVerify paper, and I'd like to understand whether the problem with floating point computation is also present in other state-of-the-art verifiers and heuristic attacks. This would be helpful for the community in understanding the extent of the vulnerability.

## Verifiers

Here are some verifiers that perform well on existing tasks and for which code is easily available. I've described why I think it's worth checking their vulnerability to the fooling approach.

- *nnenum* (CAV '20) [Code](https://github.com/stanleybak/nnenum): State of the art results (or close) on many properties.
- *ERAN* (NeurIPS '19, ICLR '19, ...) [Code](https://github.com/eth-sri/eran): State of the art results (or close) on many properties.
- *Neurify* (NeurIPS '18) [Code](https://github.com/tcwangshiqi-columbia/Neurify): A common baseline compared to in many papers.
- *ReluVal* (USENIX Security '17) [Paper](https://openreview.net/forum?id=4IwieFS44l&noteId=9OTZqSDDqdH), [Code](https://arxiv.org/pdf/1804.10829.pdf): As [AnonReviewer3 identifies](https://openreview.net/forum?id=4IwieFS44l&noteId=9OTZqSDDqdH), this paper explicitly addresses the issue of round-off, and could be robust to the fooling approach in the paper.

## Heuristic Attacks

Two strong attacks I know of for $L_\infty$ bounded perturbations are standard [projected gradient descent](https://arxiv.org/abs/1706.06083) as well as [Brendel & Bethge's attack](https://arxiv.org/pdf/1907.01003.pdf). These are both implemented in [`foolbox`](https://foolbox.readthedocs.io/en/stable/modules/attacks.html) (as well as a number of other libraries of attacks, such as `cleverhans` and `advertorch`), so they should be relatively easy to test. The relevant implementations in `foolbox` are `LInfPGD` and `LInfinityBrendelBethgeAttack`.

---

> ### Public Comment · ~Vincent_Tjeng1 · 2020-11-11
> **Additional Questions and Comments**
>
> I also had some questions as well as comments that I hope are helpful.
>
> ## Questions
> - The last two paragraphs of Section 6 mention that perturbations result in a network that still has the backdoor behavior ~50% of the time. Is this the case since the sum of the inputs to C is equally likely to be a large positive or large negative value? (I think it's worth elaborating why this is expected to be the case, since the probability that the backdoor behavior is preserved affects the number of independently sample perturbations that need to be drawn to obtain the necessary degree of confidence that there is no backdoor.)
> - Out of curiosity, for the same perturbations, is the functionality of the backdoor _across_ samples approximately 50%? (This would make an approach of verifying a single perturbed network more viable).
>
> ## Comments
> - It would help reproducibility if the paper specified the release of MIPVerify used. You should be able to determine this by the command:
>
> ```sh
> julia -e "using Pkg; Pkg.status(\"MIPVerify\")"
> ```
>
> - Figure 3 and Table 2 refer to the WK17a network, but the details of this network are not described anywhere in the paper. I'm familiar with this name as I used it in the MIPVerify paper, but I think it would be helpful to provide more details for other readers: a line explaining which network this is in the Wong & Kolter paper would be preferred, but a reference to the name in the MIPVerify paper could also work.
> - It's worth noting in the caption of Table 2 (if you have space) that the value for test accuracy is the average over 10 perturbed networks; I was initially confused how the number 0.98105 was obtained given that there were only 10000 samples in the tet set.

---

> ### Author Response · Authors · 2020-11-12
> **Initial reply**
>
> Thank you for the pointers and the ideas. We agree that it would add value if we looked at more verifiers. In the next two weeks we will do our best to try them out.
>
> The heuristic attacks are especially interesting. We did not experiment with these, as we were focused on complete verifiers. However, our construction actually provides a defense against these attacks in the sense that the backdoor is not revealed, if we apply a small modification. This is partly because in the small network (Fig 1) the implemented function is, practically speaking, a step function, so there is no gradient to rely on.
>
> The backdoor, as presented in the paper, can be detected. We already verified this experimentally: if we attack the network with PGD, the backdoor switch itself does not provide a gradient, however, the gradient of the original network tends to increase the value of the backdoor pixel (which is originally black) so eventually we end up in the backdoor input subspace (ie the subset of inputs that activate the backdoor).
>
> Also, the Brendel&Bethge attack will start in the backdoor input subspace with high probability and once inside, it is guaranteed to stay there. However, with a small modification we were able to hide the backdoor: one has to use the combination of two pixels as a backdoor pattern instead of just one, and this way the backdoor space will become hidden from both attacks. This is an unexpected finding that is interesting in itself. Our defense (perturbations) does not fix this either. We plan to elaborate on this in the final version, thank you again for suggesting to investigate this.
>
> Your question about the 50% probability in Section 6 is discussed in the Appendix where a small derivation is included that explains why and how we arrive at this 50% figure. In a nutshell, yes, it is because the noise can cause positive or negative shifts in the difference of the two “big” numbers. In addition, we plan to make this much clearer in Section 6 in the final version.
>
> Your question about the reproducibility is also probably answered in the Appendix, where we list in detail the versions of the entire software stack we used , including that of MIPVerify. We would be happy to add more details, if we missed anything that is necessary to reproduce our work, please let us know.
>
> Thank you for the remaining comments about the presentation, we will do our best to address these.

---

### Author Response · Authors · 2020-11-12
**Initial reply**

We are grateful for the very useful, detailed, and constructive comments and recommendations. Here, we shortly summarize the most important improvements that we plan to do (please also see our detailed initial replies to the individual reviews). First, we plan to improve the presentation that received lots of criticism in many of the comments and that apparently caused a number of misunderstandings. We will try to clarify these both in our answers here and in the revised manuscript as well. We also plan to extend our coverage of related work that will include additional measurements using other state-of-the-art verifiers, to the extent that the (now less than) 2-week window will allow us. When submitting the final version, we will explain in detail what we improved and added.

---

### Author Response · Authors · 2020-11-25
**Changes in rebuttal revision**

We thank everyone again for all the comments we received. Based on these comments, we were able to better formulate our main claim and we could better position the results in relation to other verifiers as well.
In a nutshell, we

- extended the related work with several items
- added a discussion about the floating point issue in verification (new section 2.4)
- performed a full reorganization of Section 6 (defense) to improve presentation (this includes new measurements with additional epsilon values)
- reorganized and clarified Section 5 as well rather significantly
- added practical motivation in the introduction with a reference to an EU report on the necessity of standard procedures to ensure safe AI
- added minor clarifications at several other places as well, following the advice we received
- added some results and discussion with other verifiers in the Appendix

Some of these points in more detail:

**Related work and the floating point angle.** We extended the related work with several items. We also included a discussion of the floating point issue in Section 2.4. However, we emphasize that the main issue is not whether one uses floating point or precise arithmetic, that is just one facet of the problem. As we describe in Section 2.4, the main issue is whether the verification matches the actual behavior of the network that is being verified. For example, the problem does not go away if one uses precise arithmetic if the network itself applies floating point, because this fact can also be exploited in principle: one simply has to make sure the network does make a roundoff error in practice while the verification will surely not make the same error. The only solution is to develop verifiers that explicitly take into account the floating point representation and that are not sensitive to the possible orderings of the operations that the network executes.

**Measurements with additional verifiers (Appendix).** We were able to complete some measurements with *reluval* and *neurify*. Reluval is not fooled by our adversarial network but it cannot deal with the WK17a variants. The neurify implementation can be fooled using the simple adversarial network as well as the backdoored WK17a. We also tested *ERAN* with *DeepPoly* on WK17a and 3% of the examples are wrongly classified as safe over the backdoored network (where all examples are known to be unsafe) and the rest of the examples all “failed” (not able to decide). We note, though, that we were not able to reproduce the results in the DeepPoly paper exactly (only approximately) so our setup might slightly differ from the one used there. Finally, we tested *Nnenum* as well. Nnenum can be fooled by a slightly modified version of our simple adversarial network, as well as the backdoored network.

**Heuristic attacks.** We also experimented with heuristic attacks (PGD and Brendel&Bethge). As we described already in our initial reply, the attacks find the backdoor in WK17a, however, with some small modifications the backdoor can be hidden from these attacks.

---

> ### Comment · ~Gagandeep_Singh1 · 2021-01-14
> **Regarding example on ERAN**
>
> Dear authors,
>
> Congratulations on the paper, I believe that your results open new directions for the design of future NN verifiers. I wanted to make sure that we avoid confusion about the applicability of the results presented here. ERAN with DeepPoly is floating-point sound meaning that if computations were performed with 64-bit according to IEEE 754 standard as specified in the source code then the verification results hold under all rounding modes and order of computations. The results about ERAN in the paper are for complete verification, obtained via the combination of DeepPoly with MILP, which we call RefinePoly. Currently, the text in the paper seems to suggest that you find counterexamples for DeepPoly which is not quite true. I hope that you can adjust the text to reflect this in the final version.
>
> Cheers,

---

> > ### Comment · ~Márk_Jelasity1 · 2021-01-19
> > **Fixing a typo**
> >
> > Thank you for pointing out the confusion with the names, we will emphasize more that we tested the combination of DeepPoly with MILP and we will use the name RefinePoly to refer to this method.

---

### Decision · Program_Chairs · 2021-01-07
**Final Decision**

**Decision:**

Accept (Poster)

**Comment:**

The authors demonstrate that complete neural network verification methods that use limited precision arithmetic can fail to detect the possibility of attacks that exploit numerical roundoff errors. They develop techniques to insert a backdoor into networks enabling such exploitation, that remains undetected by neural network verifiers and a simple defence against this particular backdoor insertion.

The paper demonstrates an important and often ignored shortcoming of neural network verification methods, getting around which remains a significant challenge. Particularly in adversarial situations, this is a significant risk and needs to be studied carefully in further work.

All reviewers were in agreement on acceptance and concerns raised were adequately addressed in the rebuttal phase, hence I recommend acceptance. However, a few clarifications raised by the official reviewers and public comments should be addressed in the final revision:
1) Acknowledging that incomplete verification methods that rely on sound overapproximation do not suffer from this shortcoming.
2) Concerns around reproducibility of MIPVerify related experiments brought up in public comments.

---

> ### Comment · ~Márk_Jelasity1 · 2021-03-02
> **final version uploaded**
>
> We uploaded the final version. About the two remaining comments: as for reproducibility, we shared our code on github (https://github.com/szegedai/nn_backdoor) and by the camera ready deadline we will also upload a docker image containing the software stack to reproduce our measurements.
>
> As for the incomplete verification methods: like in the case of complete verification, whether a given overapproximation method suffers from this shortcoming depends on whether the  implementation is mathematically provable to be sound, considering all the relevant details. In other words, it depends on whether any corners were cut for efficiency. If the implementation is sound then it is self-evident that the given implementation is not vulnerable to our attack (but see also Section 2.4 for a more nuanced discussion).